# Nanoscale magnonic Fabry-Pérot resonator for low-loss spin-wave manipulation

Huajun Qin [1✉], Rasmus B. Holländer [1], Lukáš Flajšman [1], Felix Hermann[1,2], Rouven Dreyer[3], Georg Woltersdorf[3] & Sebastiaan van Dijken [1✉]

Active control of propagating spin waves on the nanoscale is essential for beyond-CMOS magnonic computing. Here, we experimentally demonstrate reconfigurable spin-wave transport in a hybrid YIG-based material structure that operates as a Fabry-Pérot nanoresonator. The magnonic resonator is formed by a local frequency downshift of the spin-wave dispersion relation in a continuous YIG film caused by dynamic dipolar coupling to a ferromagnetic metal nanostripe. Drastic downscaling of the spin-wave wavelength within the bilayer region enables programmable control of propagating spin waves on a length scale that is only a fraction of their wavelength. Depending on the stripe width, the device structure offers full nonreciprocity, tunable spin-wave filtering, and nearly zero transmission loss at allowed frequencies. Our results provide a practical route for the implementation of low-loss YIG-based magnonic devices with controllable transport properties.

[1] NanoSpin, Department of Applied Physics, Aalto University School of Science, Aalto, Finland. [2] Physikalisches Institut, Karlsruhe Institute of Technology, Karlsruhe, Germany. [3] Institute of Physics, Martin Luther University Halle-Wittenberg, Halle, Germany. ✉email: huajun.qin@aalto.fi; sebastiaan.van.dijken@aalto.fi

A Fabry–Pérot resonator (or interferometer) is an optical cavity made of two parallel reflecting surfaces. Optical waves transmit through the resonator when incoming waves interfere constructively with circulating waves inside the cavity, whereas destructive interference produces minimal transmission[1,2]. In optics, Fabry–Pérot resonators are widely used in telecommunication networks, dichroic filters, lasers, and spectrometers to control light or measure the wavelength. Compared to electromagnetic waves, the wavelength of spin waves or magnons (quanta of spin waves) in magnetic systems is orders of magnitude smaller at GHz frequencies, facilitating the miniaturization of wave-based devices. Magnonics based on the transfer of angular momentum in the form of spin waves provides a promising technology platform for parallel information processing and tunable microwave components[3–5]. A magnonic analog to the optical Fabry–Pérot resonator, especially when operating on the nanoscale, would offer an attractive means to manipulate spin-wave transport.

Utilizing spin-wave interference inside a magnonic cavity is conceptually different from Bragg scattering on a periodic magnetic structure, known as a magnonic crystal[6–9]. Scattering of propagating spin waves in a magnonic crystal opens up forbidden frequency gaps when the wavelength ($\lambda$) matches $2a/n$ (crystal period $a$, integer $n$). If the scattering efficiency of individual units is low, robust bandgaps only form for a large number of scattering units. Since the minimal crystal period is at least half the wavelength of propagating spin waves ($n = 1$), the size of a magnonic crystal is often much larger than the spin-wave wavelength. Moreover, scattering in magnonic crystals not only suppresses the transport of spin waves within the bandgaps, but also limits the transmission signal at allowed frequencies.

Realizing a magnonic Fabry–Pérot resonator would require two parallel reflecting interfaces within a low-loss magnetic material, so that spin waves entering the magnonic cavity coherently circulate before damping out. Ferrimagnetic YIG films are the prime candidate for spin-wave transport with low loss[10–14]. Reflecting interfaces inside a continuous YIG layer can be formed, for instance, by utilizing the Oersted field of a current-carrying wire[15]. Placing patterned ferromagnets in the vicinity of a YIG film provides another attractive option. In this geometry, two interaction mechanisms can be distinguished. For static dipolar coupling, induced changes of the effective magnetic field within the YIG layer alter the propagation of spin waves[16]. Non-uniformities in the effective magnetic field can also be exploited for short-wavelength spin-wave emission[17]. Because of the static nature of the interaction in these examples, the effects extend over a broad frequency range. On the other hand, a magnetic element that is driven into resonance operates as a microwave-to-spin-wave transducer when placed near a magnetic film[18]. Dynamic dipolar coupling in this configuration produces unidirectional spin-wave excitation, which can be tailored to short wavelengths through the use of a magnetic grating[19,20]. The driven nature of this interaction locks the dynamic response of the two-magnet system to the resonance frequency of the patterned ferromagnet, limiting its operation to a narrow frequency range.

Here, we introduce a magnonic Fabry–Pérot nanoresonator allowing versatile manipulation of low-loss spin-wave transport. The resonator structure consists of a ferromagnetic metal stripe on top of a continuous nanometer-thick YIG film. Local dynamic dipolar coupling between the two magnetic layers produces two magnonic interfaces within the YIG film at the bilayer edges. At the interfaces, propagating spin waves partially reflect/transmit and their wavelength converts. Destructive interference between incoming and circulating spin waves inside the YIG/ferromagnetic metal bilayer region suppresses the transmission signal at discrete frequencies. Drastic downconversion of the spin-wave

wavelength within the narrow bilayer facilitates manipulation of micrometer-long spin waves ($\lambda = 10 - 50$ μm) by single nanostripes with widths down to 270 nm. Other attractive features of the magnonic Fabry–Pérot resonator include operation over a broad frequency range, nearly zero transmission loss at allowed frequencies, great flexibility in the design of forbidden frequency bands, and active modulation of the output signal via magnetic gating.

## Results

**Magnonic Fabry–Pérot resonator**. Figure 1a shows a schematic of the magnonic Fabry–Pérot resonator and the measurement geometry. The device structure consists of a pulsed-laser-deposited YIG film with a ferromagnetic metal stripe patterned on top. The YIG film is 70 nm or 100 nm thick and the metallic stripe is separated from the continuous YIG film by a 5-nm-thick $TaO_x$ spacer. Propagating spin waves are excited by a microwave antenna in an uncovered area of the YIG film. Spin-wave transport across the bilayer area is recorded by broadband spin-wave spectroscopy using a parallel microwave antenna at a distance of 200 μm or imaged directly by time-resolved magneto-optical Kerr effect (TR-MOKE) microscopy. We use CoFeB and permalloy (Py) as the stripe material and systematically tailor the transmission of spin waves by varying the stripe width from 270 nm to 50 μm. The Gilbert damping constant of the YIG film grown onto a (111)-oriented $Gd_3Ga_5O_{12}$ (GGG) substrate is $5 \times 10^{-4}$ (Supplementary Fig. 1).

Figure 1b compares spin-wave transmission spectra (amplitudes of $S_{12}$ and $S_{21}$ scattering parameters) measured on an uncovered 100-nm-thick YIG film (orange lines) and the same YIG film with a 730-nm-wide CoFeB stripe (blue lines). In the measurements, a +10 mT bias field saturates the YIG and CoFeB magnetization along the blue arrow shown in Fig. 1a (Damon–Eshbach (DE) transport configuration, $k \perp M$ with $M$ in-plane). For bare YIG, spin waves propagating along $+x$ ($S_{12}$) and $-x$ ($S_{21}$) produce similar signals with a gradual decrease in intensity above the ferromagnetic resonance (FMR) frequency (1.42 GHz). With the CoFeB nanostripe patterned on top, we measure a strong suppression of the $S_{12}$ signal around 1.92 GHz. The $S_{21}$ amplitude is reduced less at the same frequency. The contour plot of the $S_{12}$ scattering amplitude (Fig. 1c) depicts how the transmission of spin waves across the YIG/CoFeB bilayer varies as a function of magnetic bias field. A clear transmission gap is only visible for positive fields. The observed nonreciprocity is not related to the spin-wave excitation process[20–23], but rather a frequency-specific feature of spin-wave transport across the YIG/CoFeB bilayer. The number of transmission gaps and the gap frequency depend on the stripe width (Fig. 1d). Whereas only one transmission gap is produced by nanoscale CoFeB stripes within the antenna excitation range, spin waves are rejected at several discrete frequencies when the stripe width extends to a few micrometers (more data in Supplementary Fig. 2). Away from the transmission gaps, transport of spin waves across the YIG/CoFeB bilayer nearly matches that of the uncovered YIG film. Patterning of a CoFeB nanostripe onto YIG therefore does not produce a significant loss at allowed frequencies. From the oscillatory component in the real or imaginary part of the $S_{12}$ signal[24], we derive a spin-wave group velocity ($v_g$) of up to ~6 km/s (Supplementary Fig. 3).

The phase-resolved TR-MOKE microscopy maps of Fig. 2a, b visualize spin-wave transmission in a 100-nm-thick YIG film with a 730-nm-wide CoFeB stripe. The excitation frequency is set outside (1.76 GHz) and inside the transmission gap (1.92 GHz) at +10 mT bias field. Spin waves are excited in the YIG film at $x = 0$ μm by a microwave antenna. The waves propagate from an

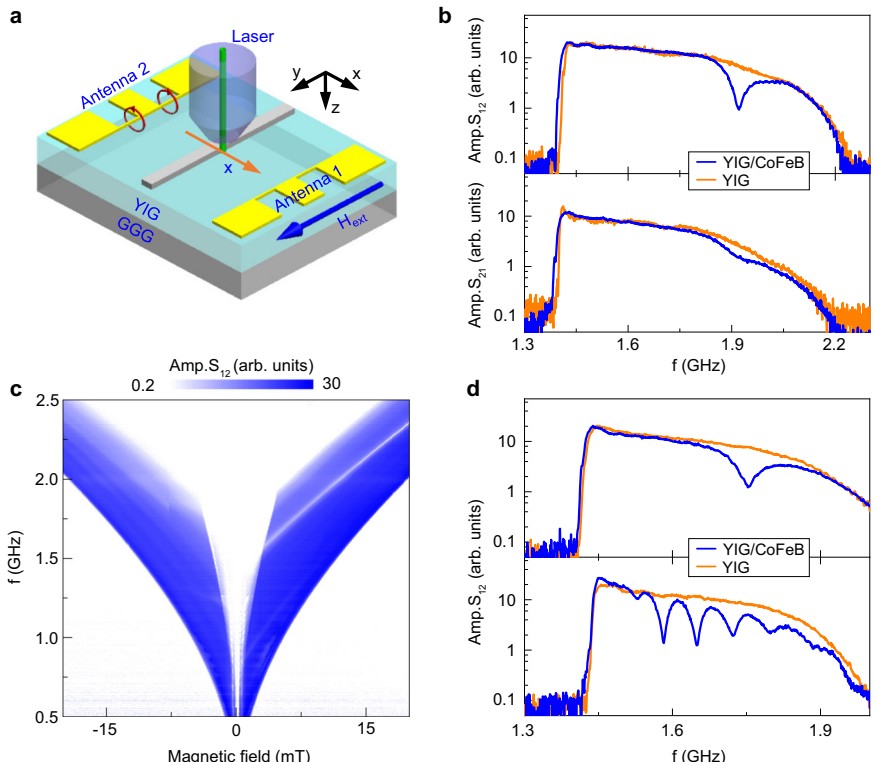

**Fig. 1 Spin-wave transmission in a YIG film with a single ferromagnetic metal stripe. a** Schematic of the experimental geometry. Spin waves are excited in a nanometer-thick YIG film using a microwave antenna and are detected by a second antenna or imaged by TR-MOKE microscopy. A magnetic bias field ($H_{ext}$) saturates the magnetization of YIG and the ferromagnetic metal parallel to the stripe. **b** Spin-wave transmission spectra (amplitude of $S_{12}$ and $S_{21}$) recorded on an uncovered 100-nm-thick YIG film (orange) and the same YIG film with a 730-nm-wide CoFeB stripe (blue). $\mu_0 H_{ext} = +10$ mT. **c** Contour plot of the $S_{12}$ amplitude as a function of magnetic field. **d** Spin-wave transmission spectra (amplitude of $S_{12}$) recorded on a 100-nm-thick YIG film with a 1.1-μm-wide and a 6.0-μm-wide CoFeB stripe (blue curves in top and bottom panel, respectively). Reference measurements on uncovered YIG films are shown in orange.

uncovered YIG film ($x < 100$ μm), across the narrow YIG/CoFeB bilayer region (indicated by the dashed line), into another uncovered YIG area ($x > 100$ μm). At 1.76 GHz, the spin waves propagate across the YIG/CoFeB bilayer almost unperturbed in the $+x$ direction (Fig. 2a). In contrast, the CoFeB nanostripe strongly suppresses the transmission of spin waves at 1.92 GHz (Fig. 2b), in agreement with the $S_{12}$ spectrum shown in Fig. 1b. Complementary TR-MOKE maps for other excitation frequencies and magnetic bias fields are shown in Supplementary Fig. 4. Micromagnetic simulations performed in MuMax3[25] corroborate the experimental observations (Fig. 2c, d). The simulated time evolution of spin-wave transport demonstrates that the CoFeB nanostripe does not cause any disturbance at 1.76 GHz. Yet, at 1.92 GHz, a clear interference pattern forms in the YIG film ahead of the YIG/CoFeB bilayer ($x < 100$ μm), signifying strong spin-wave reflection and a resultant loss in transmission.

Importantly, the simulations reveal also that the incoming spin waves convert to a propagating mode with much shorter wavelength within the YIG/CoFeB bilayer region (see inset in Fig. 2c). After transport across the bilayer, the spin-wave wavelength reverts back to its original value in the second uncovered area of the YIG film. The data shown in Fig. 2e–h for a CoFeB stripe width of 30 μm shed more light on the mode conversions. For waves propagating along $+x$ and a bias field of $+10$ mT (Fig. 2e, g), the spin-wave wavelength converts down drastically in the YIG/CoFeB bilayer (the resolution of TR-MOKE microscopy is insufficient to resolve this mode). Because the decay length of the short-wavelength spin waves is small, no signal is measured in the YIG film beyond the CoFeB stripe. When the

propagation direction is reversed (Fig. 2f, h), the wavelength of incoming spin waves again downconverts in the YIG/CoFeB bilayer. However, the wavelength of this mode is not equally short and it damps out less quickly. Consequently, spin waves propagating along $-x$ are not completely blocked by the CoFeB stripe. Full nonreciprocity demonstrated here extends over a broad frequency range for 25–50-μm-wide stripes (Supplementary Fig. 5). In structures with narrower CoFeB stripes, the different damping characteristics of counter-propagating spin waves in the YIG/CoFeB bilayer affect the depth of the transmission gap (Fig. 1b). Hereafter, we number the three types of spin-wave modes in our system as 1 (uncovered YIG), 2 (YIG/CoFeB bilayer, long wavelength), 3 (YIG/CoFeB bilayer, short wavelength), i.e., $\lambda_1 > \lambda_2 \gg \lambda_3$. From fits to experimental spin-wave profiles (Supplementary Fig. 6), we derive a decay length of $l_d > 400$ μm for the $\lambda_1$ mode and $l_d > 30$ μm for $\lambda_2$ spin waves at 1.7–1.9 GHz and $+10$ mT. The wave profile of the short-wavelength $\lambda_3$ mode is not resolved by TR-MOKE microscopy. Instead, we estimate a decay length of $l_d = 10$ μm for $\lambda_3$ spin waves by fitting the dependence of the nonreciprocity coefficient on CoFeB stripe width (Supplementary Fig. 5). The drastic downconversion of the spin-wave wavelength from $\lambda_1 = 12.8$ μm to $\lambda_3 = 310$ nm in the YIG/CoFeB bilayer (data derived at 1.76 GHz (Fig. 2e, g)) offers an attractive mechanism for short-wavelength spin-wave excitation, complementing other methods such as the use of spin-torque nano-oscillators[26,27] or nano-grating couplers[28,29]. Next, we will demonstrate that the transmission gaps measured on structures with narrow CoFeB stripes are caused by destructive interference between the spin waves entering and the spin waves circulating the

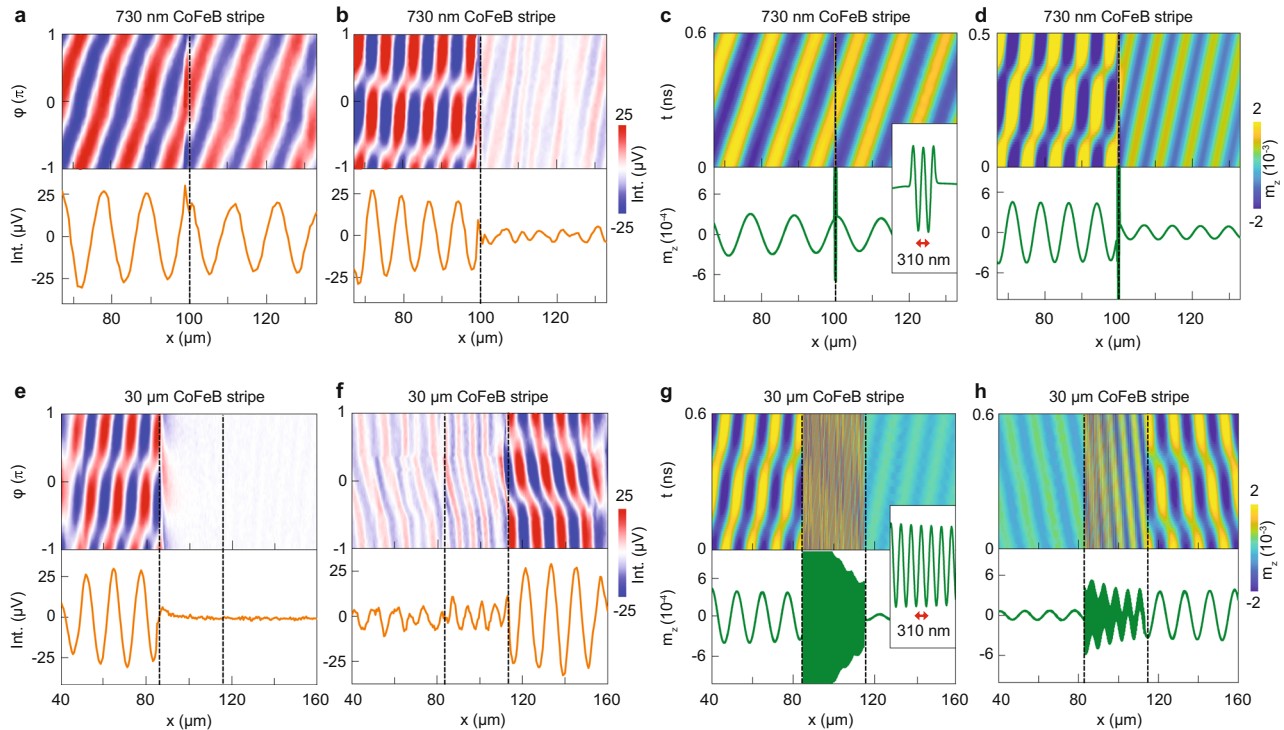

**Fig. 2 TR-MOKE microscopy of propagating spin waves in a YIG film with a single ferromagnetic metal stripe.** Phase-resolved TR-MOKE microscopy maps and line profiles measured on a 100-nm-thick YIG film with a 730-nm-wide CoFeB stripe (**a**, **b**) and corresponding micromagnetic simulations (**c**, **d**). The excitation frequency is 1.76 GHz (**a**, **c**) and 1.92 GHz (**b**, **d**) and $\mu_0 H_{ext} = +10$ mT. The inset in (**c**) zooms in on the YIG/CoFeB bilayer. **e**, **f** Phase-resolved TR-MOKE microscopy maps and line profiles recorded on a 100-nm-thick YIG film with a 30-μm-wide CoFeB stripe for spin waves propagating along $+x$ (**e**) and $-x$ (**f**). **g**, **h** Corresponding micromagnetic simulations. The excitation frequency in **e**–**h** is 1.76 GHz and $\mu_0 H_{ext} = +10$ mT. Depending on the propagation direction, the incoming $\lambda_1$ spin wave converts to a $\lambda_3$ wave (**e**, **g**) or a $\lambda_2$ wave (**f**, **h**). The dashed lines in the graphs mark the CoFeB stripe.

YIG/CoFeB bilayer, analogous to the operation of an optical Fabry–Pérot resonator.

**Spin-wave dispersion, mode conversion, and transmission gap formation.** As a starting point, we summarize the measured and simulated spin-wave dispersion relations for the three different modes in our system (solid and open symbols in Fig. 3a). In addition, we use Peter Grünberg's model[30] to calculate the dispersion of propagating spin waves in the YIG/CoFeB bilayer. To account for the dipole-exchange character of the short-wavelength $\lambda_3$ mode, we added exchange terms for the individual CoFeB and YIG layers to the model by substituting $H \rightarrow H_{ext} + 2A/(\mu_0 M_s)k_z^2$ (see "Methods" for details). The blue curve in Fig. 3a depicts the calculation result. The dispersion relation of the bilayer is fully determined by dynamic dipolar coupling between the CoFeB and YIG layers[31], as interlayer exchange coupling across the 5-nm-thick $TaO_x$ spacer is assumed to be zero. The orange curve in Fig. 3a shows the spin-wave dispersion in the uncovered YIG film, as calculated using the expression of Kalinikos and Slavin[32]. The experimental, simulated, and calculated data agree well and complement each other. The results of Fig. 3a demonstrate an asymmetric spin-wave dispersion in the YIG/CoFeB bilayer. For a magnetic bias field of $+10$ mT, the dispersion relation of the bilayer shifts down slightly compared to that of uncovered YIG for $-k$. The dispersion curve for $+k$ is affected much more, displaying a stronger frequency downshift and a negative group velocity for small $k$. Propagating modes with differing wavelengths ($\lambda_1$ for both $-k$ and $+k$, $\lambda_2$ for only $-k$, and $\lambda_3$ for only $+k$) are available at the same frequency in the uncovered YIG and YIG/CoFeB bilayer regions, facilitating wavelength conversions over a broad frequency range. The two

edges of the YIG/CoFeB bilayer where the spin-wave dispersion changes, act as the interfaces of the magnonic Fabry–Pérot resonator.

Using the asymmetrical dispersion relation of the YIG/CoFeB bilayer, we derive a condition for the formation of spin-wave transmission gaps. If propagating spin waves within the bilayer accumulate a total phase of $(2n + 1) \times \pi$ upon two subsequent internal reflections within the bilayer, incoming and circulating spin waves interfere destructively. As a result, gaps in the transmission spectrum form at discrete frequencies. Because spin waves traverse the YIG/CoFeB bilayer once with wavelength $\lambda_2$ and once with $\lambda_3$ during an internal reflection cycle, the resonance condition producing minimal transmission can be written as $(|k_2| + |k_3|)w + \varphi_0 = (2n + 1) \times \pi$ (see "Methods"). In this expression, $w$ is the stripe width, $n$ is the resonance order, and $\varphi_0$ accounts for the total phase change caused by two internal reflections, effects of long-range dipolar fields, and a deviation of the effective resonator width from $w$. We note that the resonance condition does not depend on the reflection and transmission coefficients of the two resonator interfaces nor the direction of wave propagation. The amplitude of transmitted spin waves at the gap frequency, however, relies on whether the incoming $\lambda_1$ spin waves convert to the $\lambda_2$ or $\lambda_3$ mode when entering the bilayer. Our resonator model explains this nonreciprocal resonance effect by different transmission coefficients in the expressions describing spin-wave transport across the bilayer region (Eqs. (5) and (9) in "Methods") and the fact that $\lambda_3$ waves decay faster than $\lambda_2$ waves.

We use micromagnetic simulations to study spin-wave transmission across the two interfaces of the resonator structure (Supplementary Fig. 7). Multiple internal reflections are avoided in this analysis by using a 50-μm-wide CoFeB stripe. For spin

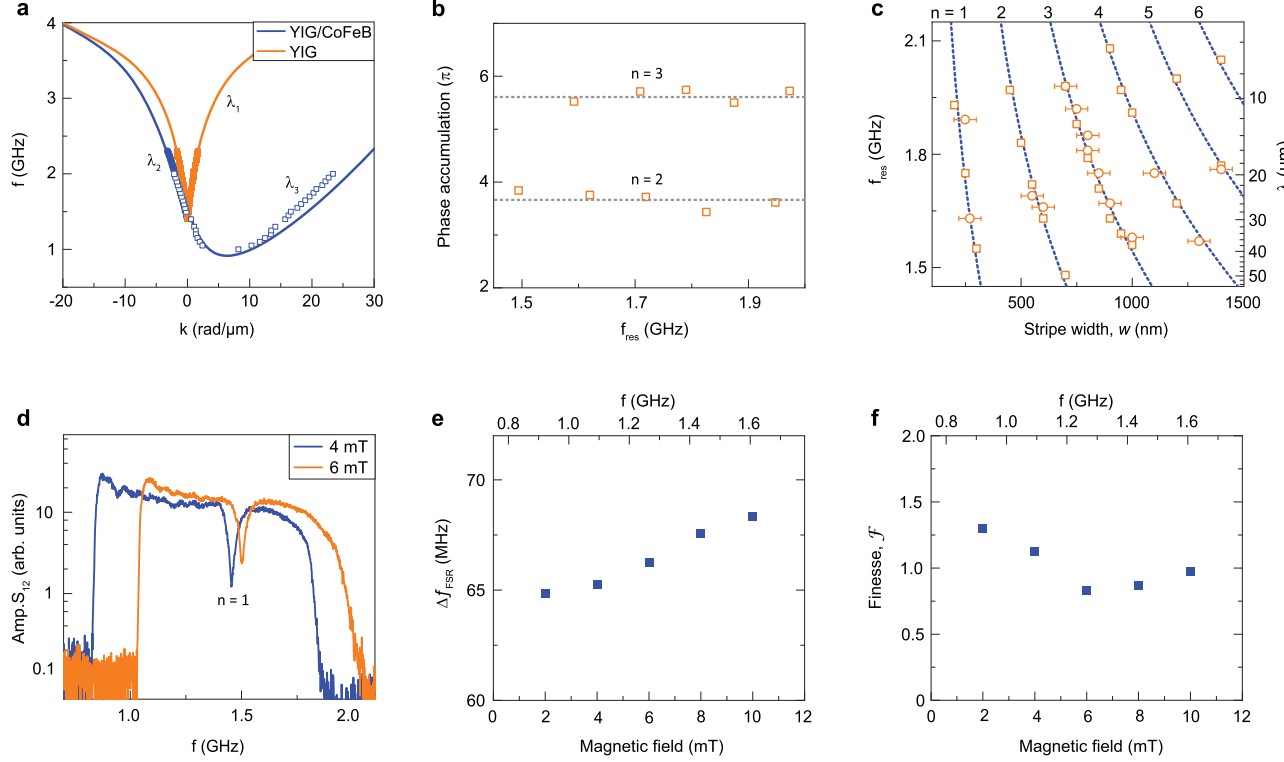

**Fig. 3 Dispersion relations and resonant scattering of spin waves. a** Experimental (solid symbols), simulated (open symbols), and calculated (lines) spin-wave dispersion relations for a 100-nm-thick YIG film ($\lambda_1$) and a YIG/CoFeB bilayer ($\lambda_2$ and $\lambda_3$). $\mu_0 H_{ext} = +10$ mT and the thickness of CoFeB is 50 nm. **b** Simulated phase accumulation during a full internal reflection cycle for the $n = 2$ and $n = 3$ transmission gaps for different stripe widths. Parameter $\varphi_0$ is extracted as the difference between $5\pi$ ($n = 2$) and $7\pi$ ($n = 3$) and the average value of the simulated phase accumulations of these modes (dashed lines). **c** Frequency of the spin-wave transmission gap as a function of CoFeB stripe width for $n = 1$–6. The blue dashed lines are calculated using the condition of minimum transmission and the asymmetric dispersion relation of the YIG/CoFeB bilayer (squares in (**a**)). A constant value of $\varphi_0 = 1.34\pi$ is used for all transmission gaps. Experimental results (circles) and simulations (squares) of transmission gap frequencies agree with the magnonic resonator model. **d** Spin-wave transmission spectra (amplitude of $S_{12}$) recorded on a 100-nm-thick YIG film with a 270-nm-wide CoFeB stripe. $\mu_0 H_{ext} = +4$ mT and +6 mT. The transmission gap in the spectrum corresponds to the $n = 1$ resonance condition. **e**, **f** Free spectral range ($\Delta f_{FSR}$) and finesse ($\mathcal{F}$) of a 6-μm-wide magnonic Fabry–Pérot resonator as a function of magnetic bias field and frequency.

waves converting from the $\lambda_1$ to the $\lambda_2$ mode (incoming waves) or vice versa (outgoing waves), we estimate transmission coefficients $t_{12} = 1$ and $t_{21} = 0.5$, respectively. The quantitative assessment of transmissions involving $\lambda_1$ and $\lambda_3$ modes is complicated by a difference in wave localization and spin-wave ellipticity ($\epsilon = 1 - m_{min}/m_{max}$)[33]. While the wave profiles of the $\lambda_1$ and $\lambda_2$ modes are approximately uniform across the YIG film thickness, the short-wavelength $\lambda_3$ mode localizes at the top surface. Also, $\lambda_1$ and $\lambda_2$ waves are elliptical along $x$ ($\epsilon = 0.8$), whereas the $\lambda_3$ mode near the top surface is elliptical along $z$ ($\epsilon = 0.3$). Efficient transmission of spin waves across single interfaces for some of the mode conversions may be attributed to backscattering immunity, an effect that is prominent for DE spin waves inside the volume-mode gap[34,35]. We note that stronger spin-wave reflection at other interfaces does not contradict low-loss spin-wave transmission across nanoresonators at allowed frequencies (Figs. 1 and 2). Analogous to an optical Fabry–Pérot resonator, multiple internal reflections and constructive interference between incoming and circulating spin waves produces high transmission even if the reflection coefficients of the two interfaces are non-zero. This conclusion is supported by a comparison of phase-resolved TR-MOKE microscopy maps and model calculations using incoming, reflected, and outgoing spin waves (Supplementary Fig. 8). For a 730-nm-wide resonator, we find a total transmission $T = 0.92$ at an allowed frequency of 1.76 GHz. Moreover, because the experimental data and calculations only agree when the

absorption coefficient is set close to zero, the energy loss caused by dipolar coupling to CoFeB is small in the nanoscale resonator.

Next, we use the destructive interference condition to derive the frequency of spin-wave transmission gaps. The total phase accumulation within the YIG/CoFeB bilayer ($\varphi_0$) at the gap frequency is estimated from micromagnetic simulations. Figure 3b shows results for $n = 2$ and $n = 3$ obtained by combining data for different stripe widths. From this graph, we extract $\varphi_0 = 1.34\pi$ for both $n = 2$ and $n = 3$, irrespective of frequency. We use this parameter, the asymmetric dispersion relation of the bilayer (Fig. 3a), and the resonance condition to map out the frequencies of spin-wave transmission gaps as a function of CoFeB stripe width. The dashed blue lines in Fig. 3c depict the results for $n = 1$–6 (Supplementary Fig. 9 shows an extended version of this graph with data up to $n = 18$). In addition, we plot the extracted gap frequencies from our experiments (circles) and simulations (squares) on more than 20 samples. Good agreement between the data sets validates the modified Fabry–Pérot model with differing counter-propagating spin-wave modes. We note that the reason for this correspondence is not a priori obvious, as long-range dipolar interactions are expected to alter the physical properties of the magnonic resonator[15]. Our experiments and simulations confirm this. For instance, dipolar coupling fields modify the wavelength of spin-wave modes near the bilayer edges and produce a frequency shift between the destructive interference conditions within the bilayer region and the transmission gaps (see "Methods" and Supplementary Fig. 10). However, as

demonstrated here, a constant value of $\varphi_0$ effectively accounts for both effects. Because $k_3 >> k_2$, the destructive interference condition is mostly determined by the short-wavelength $\lambda_3$ mode, enabling extensive downscaling of the single-stripe device. For the $n = 1$ resonance, incoming spin waves with a wavelength ranging from 50 to 10 μm (right axis of Fig. 3c) are manipulated by 300–200 nm wide CoFeB stripes, a feature that could never be accomplished by a magnonic crystal. Figure 3d experimentally demonstrates the formation of a $n = 1$ transmission gap for a CoFeB stripe width of only 270 nm.

In analogy to an optical Fabry–Pérot resonator[36], we evaluate the free spectral range ($\Delta f_{FSR}$) and finesse ($\mathcal{F}$) of our magnonic structure. Because of conversions between $\lambda_3$ and $\lambda_2$ modes, the round-trip time ($t_{RT}$) of spin waves circulating the resonator is given by $w/v_g(\lambda_3) + w/v_g(\lambda_2)$. Using the definitions $\Delta f_{FSR} = 1/t_{RT}$ and $\mathcal{F} = \Delta f_{FSR}/\Delta f$, with $\Delta f$ the Lorentzian linewidth of the resonance peak between two transmission gaps, we derive both parameters as a function of frequency and magnetic bias field from experimental transmission spectra on a 6-μm-wide resonator. The results are depicted in Fig. 3e, f and more details are provided in Supplementary Fig. 11. Compared to its optical counterpart, the magnonic Fabry–Pérot resonator offers several attractive features including drastic downscaling of the device size at GHz frequencies and nonreciprocal signal transport. Magnonic Fabry–Pérot resonators with optimized finesse values could therefore find applications in narrowband microwave filtering and spectroscopy.

**Transmission gap design**. Destructive interference of spin waves in a YIG film with a dipolar-coupled ferromagnetic metal stripe

provides versatility in the design of magnonic transport properties. Our simple resonator structure has several compelling attributes. First, by employing a continuous YIG film, potential detrimental effects caused by nanopatterning of YIG edges perpendicular to the spin-wave propagation direction are avoided. Second, significant downconversion of the spin-wave wavelength within the bilayer region enables manipulation of propagating spin waves on length scales that are much smaller than their wavelength. Third, patterning of a nanoscale CoFeB stripe on top of a YIG film results in hardly any additional transmission loss at allowed frequencies (Fig. 1b, d). We will now discuss ways to widen and deepen the transmission gap and to actively manipulate spin-wave transport in the magnonic Fabry–Pérot resonator.

One method to reduce the transmission signal at the gap frequency involves the use of multiple CoFeB stripes. Figure 4a shows spin-wave transmission spectra for devices that comprise of two and four 250-nm-wide stripes with period $p = 500$ nm on top of a 70-nm-thick YIG film. Compared to structures with only one CoFeB nanostripe (Fig. 3d and Supplementary Fig. 12), the transmission gap is wider and deeper. Figure 4b summarizes the dependence of the gap size on magnetic bias field for the device with four CoFeB stripes (solid symbols). The effect of destructive interference decreases with increasing bias field because it shortens the spin-wave decay length[24], thereby limiting the number of internal reflections within the four resonators and the resulting gap size. The transport properties of devices with multiple stripes are determined solely by the resonance properties of the individual CoFeB stripes rather than Bragg scattering, which would produce an $n = 1$ bandgap at 2.8 GHz for $p = 500$

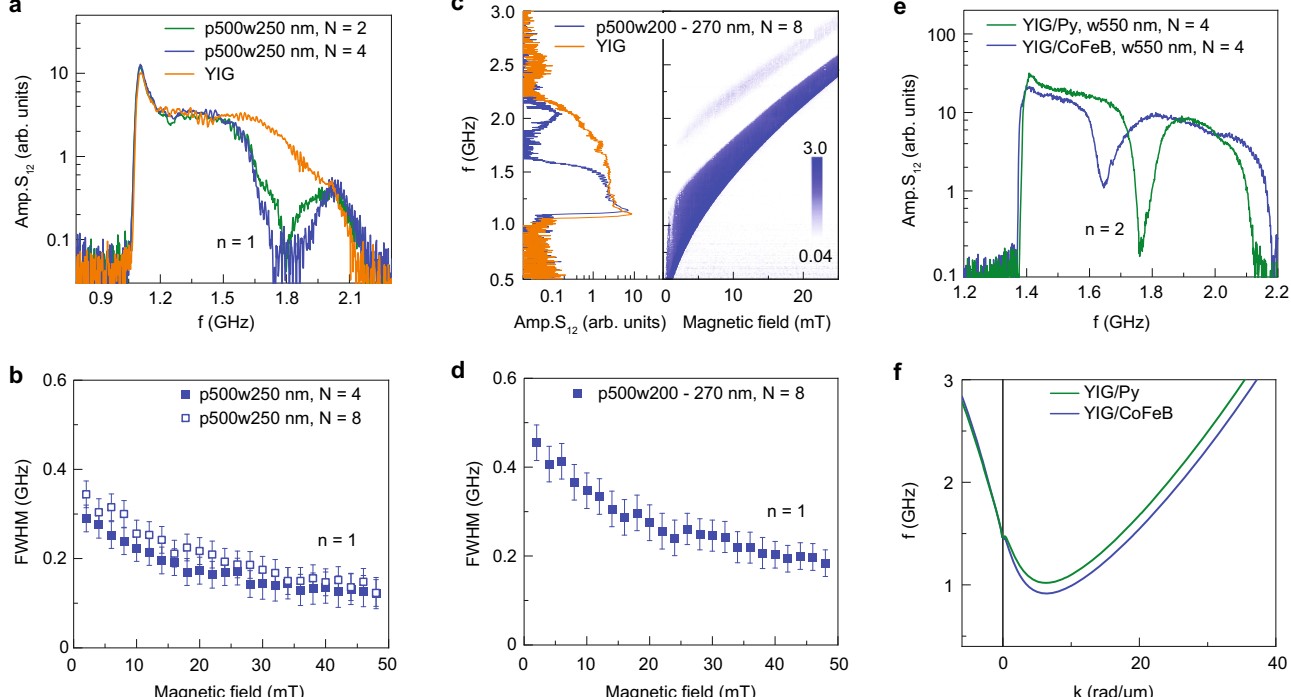

**Fig. 4 Tailoring of transmission gaps. a** Spin-wave transmission spectra (amplitude of $S_{12}$) measured on a 70-nm-thick YIG film with two and four 250-nm-wide CoFeB stripes (green and blue curves) and the same YIG film without stripes (orange curve). $\mu_0 H_{ext} = +6$ mT and the center-to-center distance between the stripes (period $p$) is 500 nm. **b** Dependence of the transmission gap size on magnetic bias field for samples with four and eight 250-nm-wide CoFeB stripes. **c** Spin-wave transmission spectra recorded at $\mu_0 H_{ext} = +6$ mT and contour plot of the $S_{12}$ amplitude as a function of magnetic bias field for a resonator structure with eight CoFeB stripes. The stripe widths increase from 200 to 270 nm in 10 nm steps. The orange curve depicts a reference measurement on the same YIG film without stripes. **d** Variation of the transmission gap size with magnetic bias field for the same sample as in (**c**). **e** Comparison of spin-wave transmission spectra recorded on resonator structures with four 550-nm-wide Py stripes and four 550-nm-wide CoFeB stripes. The YIG film is 100-nm thick, $p = 1$ μm, and $\mu_0 H_{ext} = +10$ mT. **f** Spin-wave dispersion relations for YIG/Py and YIG/CoFeB bilayers at $\mu_0 H_{ext} = +10$ mT.

nm ($k_{Bragg} = \pi/p$). Increasing the number of nanostripes to eight widens the transmission gap further (Fig. 4b). Instead of adding identical nanostripes, the transmission gap is tailored even more by a variation of the stripe width. An example of this approach for eight CoFeB stripes with widths increasing from 200 to 270 nm in 10 nm steps is shown in Fig. 4c, d. In this sample, the spin-wave signal inside the $n = 1$ transmission gap is suppressed all the way down to the background measurement level. The eight Fabry–Pérot resonators reject incoming spin waves at slightly different frequencies and, together, their filtering properties combine into a transmission gap with a size up to 0.45 GHz. The opening of gaps in this fashion can be extended easily, offering greater flexibility in the design of band structures than YIG-based magnonic crystals in terms of the gap size and depth. Moreover, the intensity of spin-wave signals at allowed frequencies is nearly the same as those of the bare YIG film, signifying truly low-loss manipulation of spin-wave transmission.

Wavelength conversions and internal reflections in the YIG film of our resonator structure are induced by dynamic dipolar coupling to a ferromagnetic metal stripe. Replacing CoFeB with a material exhibiting another saturation magnetization should therefore alter the transmission characteristics. To verify this assertion, we performed spin-wave spectroscopy measurements on a 100-nm-thick YIG film with four 550-nm-wide permalloy (Py) stripes (green curve in Fig. 4e). Compared to the same structure made of CoFeB (blue curve in Fig. 4e), a similar but shifted $n = 2$ transmission gap is measured. The increase in gap frequency is explained by an upward shift of the dispersion relation for the $\lambda_3$ mode in the YIG/Py bilayer (Fig. 4f).

Supplementary Fig. 13 provides more details on the dependence of the bilayer dispersion curve on stripe material and YIG film thickness.

**Active control of spin-wave transport.** Magnonic wave-based computing technology requires active manipulation of propagating spin waves. Reconfigurable spin-wave transport has been realized in magnonic crystals by switching the magnetization of stripe arrays between parallel and antiparallel states[37]. Doubling of the effective lattice constant explains this effect. In our magnonic Fabry–Pérot resonator, independent magnetization switching in the YIG film and the ferromagnetic metal stripe affects the dynamic dipolar coupling between the two materials. Calculations of the dispersion relation show a clear difference for parallel and antiparallel magnetization configurations (Fig. 5a). In particular, the magnitude of $k_3$ at a given frequency is larger when the YIG and CoFeB magnetization align antiparallel. Because $k_3$ mostly determines the resonance condition, a switching-induced change of this wave vector shifts the frequency of the transmission gaps. Spin-wave spectroscopy measurements in Fig. 5b, c illustrate this effect. The two panels show data for four 250-nm-wide CoFeB stripes on a 70-nm-thick YIG film. In the experiments, the magnetization of YIG and CoFeB are first saturated in a +25 mT magnetic bias field. At +10 mT, this parallel magnetization configuration produces a wide and deep $n = 1$ transmission gap at 1.92 GHz. From the dispersion curve in Fig. 5a, we estimate that $k_3 = 24.4$ rad/μm at the gap center. Sweeping the field to −10 mT reverses the direction of magnetization in the YIG film, but not in the CoFeB nanostripes (Supplementary

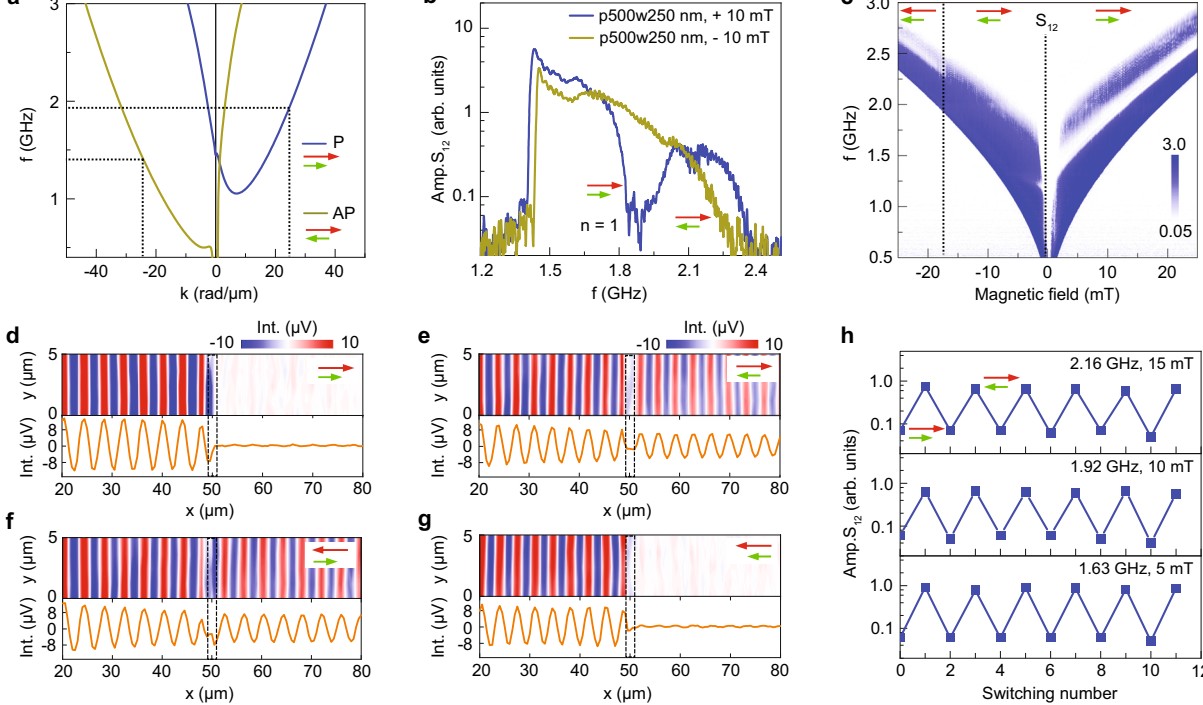

**Fig. 5 Active control of spin-wave transmission spectra. a** Calculated spin-wave dispersion relations for a 70 nm YIG/50 nm CoFeB bilayer with parallel (P) and antiparallel (AP) magnetization configurations. Switching between the two states involves magnetization reversal in the YIG film, which changes the sign and magnitude of $k_2$ and $k_3$. **b** Spin-wave transmission spectra (amplitude of $S_{12}$) of a 70-nm-thick YIG film with four 250-nm-wide CoFeB stripes ($p = 500$ nm) for parallel (blue curve, $\mu_0 H_{ext} = +10$ mT) and antiparallel (dark yellow curve, $\mu_0 H_{ext} = −10$ mT) magnetization states. The magnetization of the YIG film is switched. **c** Contour plot of the $S_{12}$ amplitude as a function of magnetic field for the same device structure. The field is swept from +25 to −25 mT and switching between different magnetization states is indicated by dotted lines. **d–g** TR-MOKE microscopy maps and line profiles recorded at 1.92 GHz for four different magnetization configurations. Dashed lines mark the region with four 250-nm-wide CoFeB stripes. **h** Modulation of the spin-wave transmission signal during sequential switching between parallel and antiparallel magnetization states. Data for three different bias fields are shown. In all panels, the red and green arrows depict the direction of magnetization in CoFeB and YIG, respectively.

Fig. 14 shows hysteresis curves for the YIG film and CoFeB stripes). In the newly established antiparallel magnetization state, the gap frequency is downshifted to about 1.4 GHz (see dotted line at $k_3 = -24.4$ rad/μm in Fig. 5a). Because this frequency is below the FMR of YIG, the transmission gap does not show up in the $S_{12}$ spectrum shown in Fig. 5b (dark yellow curve). The switching-induced downshift of the gap frequency enhances the spin-wave signal most around the initial transmission gap (1.92 GHz in this example). Since the magnetic bias field tunes the gap frequency from 1.4 to 2.3 GHz below the switching field of the CoFeB stripe (see Fig. 5c), strong signal modulations are easily reconfigured by magnetic gating within this broad frequency band. Figure 5d–g shows TR-MOKE microscopy maps of propagating spin waves at 1.92 GHz for four different magnetization configurations. The data visualize active suppression of spin-wave transmission at a fixed frequency when the magnetization alignment is switched from parallel to antiparallel. Figure 5h demonstrates the reproducibility of signal modulation via magnetic switching. At all three bias fields, the $S_{12}$ amplitude changes by at least one order of magnitude. Even larger effects can be attained using the design strategies discussed in the previous section. Supplementary Fig. 15 shows similar data recorded on a 100-nm-thick YIG film with four 550-nm-wide CoFeB stripes. In addition, Supplementary Fig. 16 depicts how the frequency of the transmission gap changes when a 10 mT bias field rotates the magnetization of the YIG film while the CoFeB magnetization remains fixed.

In summary, we have experimentally demonstrated a magnonic Fabry–Pérot resonator for low-loss spin-wave manipulation on the nanoscale. Our approach involving dynamic dipolar coupling between a narrow ferromagnetic metal stripe and a YIG film enables versatile tuning of spin-wave transport by variation of the stripe width or active gating via magnetic switching in the bilayer. Fabrication of the magnonic resonator is compatible with other YIG-based magnonic elements and provides a simple route for the integration of small-footprint magnonic devices without the need for ultrashort wavelength excitation and detection.

## Methods

**Sample fabrication**. We grew YIG films with a thickness of 70 and 100 nm on GGG(111) substrates using PLD. The GGG substrates were ultrasonically cleaned in acetone and isopropanol before loading into the deposition chamber. We degassed the substrates at 550 °C for 15 min. After this, oxygen was inserted into the chamber and the temperature was raised to 800 °C at a rate of 5 °C per minute. YIG films were deposited from a stoichiometric target in an oxygen partial pressure of 0.13 mbar at this temperature. We used an excimer laser with a pulse repetition rate of 2 Hz and a laser fluence of 1.8 J/cm². Following film growth, we first annealed the YIG films at 730 °C for 10 min in an oxygen environment of 13 mbar and then cooled them down to room temperature at a rate of −3 °C/min. The deposition process resulted in single-crystal YIG films, as confirmed by x-ray diffraction and transmission electron microscopy measurements. Ferromagnetic stripes (CoFeB, Py) with a thickness of 50 nm were patterned onto the YIG films by photolithography or electron-beam lithography, depending on stripe width. The ferromagnetic metals were grown by magnetron sputtering at room temperature onto a 5-nm-thick TaO$_x$ spacer. After growth, we performed lift-off by placing the samples in a bath of acetone. For spin-wave characterization, two parallel microwave antennas with a separation of 200 μm were patterned on top of the YIG films using a laserwriter LW405 system and magnetron sputtering. The antennas consisted of 3 nm Ta and 120 nm Au. The antenna width ($w_a$) was 2 or 6 μm, allowing the excitation of spin waves with wave vectors up to $\pi/w_a$.

**Broadband spin-wave spectroscopy**. The setup for spin-wave characterization consisted of a two-port vector network analyzer (Agilent N5222A) and a home-built electromagnet probing station. FMR spectra were recorded by placing the sample face-down onto a coplanar waveguide with a 150-μm-wide signal line. We measured spin-wave transmission spectra by measuring $S_{12}$ and $S_{21}$ scattering parameters. To avoid nonlinear excitations, the power of the microwave excitation signal was set to −10 dBm. We used a frequency sweep method to record transmission spectra at different magnetic bias fields. The strength of the magnetic field was changed in a stepwise fashion from positive to negative, unless otherwise stated. To improve contrast, we subtracted a reference spectrum taken at 100 mT

from the real and imaginary values of the $S_{12}$ and $S_{21}$ scattering parameters. The recorded real and imaginary data sets were subsequently used to calculate the amplitudes of $S_{12}$ and $S_{21}$.

**Time-resolved magneto-optical Kerr effect microscopy**. We used a home-built TR-MOKE microscope and a super-Nyquist sampling magneto-optical Kerr effect (SNS-MOKE) microscope to image the transport of spin waves in a phase-resolved manner. In TR-MOKE measurements, the RF excitation frequency was fixed to multiples of the 80 MHz laser repetition rate. The TR-MOKE signal was extracted from the measured Kerr rotation utilizing amplitude modulation of the RF signal and subsequent lock-in demodulation. The femtosecond laser was focused onto the sample by a 20× objective with a numerical aperture of 0.5, providing a spatial resolution of ~500 nm. In SNS-MOKE, the laser frequency comb downconverts the excited GHz magnetization dynamics to an intermediate frequency $\epsilon$, allowing tuning of the excitation signal to any frequency $f_{exc} = n \times f_{rep} + \epsilon$. At non-zero $\epsilon$ and with the excitation synchronized to a laser repetition rate of 80 MHz, this downconversion occurs coherently, i.e., the phase of spin waves relative to the excitation signal is preserved by lock-in demodulation at $\epsilon$[38]. The focused laser-beam spot in SNS-MOKE had a diameter of about 500 nm. In both setups, the samples were mounted on a piezo-stage and spin waves were excited using one of the microwave antennas. Spin-wave transport was spatially imaged at different RF frequencies and magnetic bias fields by moving the piezo-stage along the direction of propagating spin waves.

**Calculation of bilayer dispersion curves**. More details on the approach can be found in ref. [30]. In order to find the spin-wave dispersion relation of the bilayer system, we assume that the magnetostatic potential $\psi$ inside and outside the ferromagnetic layers is of the form $\psi_{in,ext}(x,z) = \zeta_{in,ext}(z)e^{(ik_x x)}$ and

$$\zeta_{ext}(z) = \begin{cases} Ce^{(-k_z z)} & \text{if } z > \frac{d_0}{2} + d_1 \\ De^{(k_z z)} & \text{if } z < -\frac{d_0}{2} - d_2 \\ Ee^{(k_z z)} + Fe^{(-k_z z)} & \text{if } \frac{d_0}{2} > z > -\frac{d_0}{2} \end{cases} \quad (1)$$

$$\zeta_{in}(z) = \begin{cases} G_1 e^{(k_z z)} + H_1 e^{(-k_z z)} & \text{if } \frac{d_0}{2} \leq z \leq \frac{d_0}{2} + d_1 \\ G_2 e^{(k_z z)} + H_2 e^{(-k_z z)} & \text{if } \frac{d_0}{2} \geq z \geq -\frac{d_0}{2} - d_2. \end{cases} \quad (2)$$

Here, $z$ denotes the coordinate along the out-of-plane direction and $x$ is the propagation direction, $k_x, k_z$ are the corresponding wave vectors along those directions, $d_0$ marks the spacer between layer 1 and 2, and $d_1, d_2$ are the thicknesses of layer 1 and layer 2. We then solve the following conditions at the four boundaries:

$$\psi_{in}(x,z) = \psi_{ext}(x,z) \quad (3)$$

$$(1 + \kappa_m)\frac{\partial \psi_{in}}{\partial z} - i\nu_m \frac{\partial \psi_{in}}{\partial x} = \frac{\partial \psi_{ext}}{\partial z} \quad (4)$$

By doing this, the coefficients $C$, $D$, $E$, and $F$ are eliminated from the system of equations. In order to have a solution for $G_1$, $G_2$, $H_1$, and $H_2$, the determinant of the system of equations must vanish, leading to an equation for $k$ and $\omega$. Furthermore, we use $k_x = \pm k_z$. Moreover, $\kappa_m$ and $\nu_m$ are specified as $\kappa_m = \Omega_{Hm}/(\Omega_{Hm}^2 - \Omega_m^2)$ and $\nu_m = \Omega_m/(\Omega_{Hm}^2 - \Omega_m^2)$, where $\Omega_m = \omega/(\gamma\mu_0 M_{sm})$ and $\Omega_{Hm} = H/(M_{sm})$. Here, $\omega = 2\pi f$ is the angular frequency, $\gamma$ is the gyromagnetic ratio, $\mu_0$ is the permeability of vacuum, $M_{sm}$ is the saturation magnetization of layer $m = 1, 2$, and $H$ is the magnetic field. At this point, we make the substitution $H \rightarrow H_{ext} + 2A_m/(\mu_0 M_{sm})k_z^2$ to include exchange interactions inside layer 1 and layer 2. Here, $A_m$ is the exchange constant of layer $m = 1, 2$ and $H_{ext}$ is the external magnetic bias field. This substitution corresponds to the use of an effective field in each layer. Static contributions to this term such as magnetic anisotropy could be implemented in a similar fashion, but turned out to be unnecessary for the materials studied here. The equations are then solved for $\omega$.

**Derivation of the resonance condition**. We derive an expression for destructive spin-wave interference in the YIG/CoFeB bilayer by using the region's edges within the YIG film as interfaces. The reflection and transmission coefficients of the two interfaces are labeled $r_{ij}$ and $t_{ij}$, where $i$ and $j$ indicate the spin-wave mode before and after reflection or transmission. For instance, $r_{23}$ indicates an internal reflection with a $\lambda_2 \rightarrow \lambda_3$ mode conversion. A schematic of the bilayer structure with reflection and transmission coefficients is shown in Supplementary Fig. 17. Because of the asymmetric dispersion relation in the bilayer region and differences in the wave localization and spin-wave ellipticity (Supplementary Fig. 7), $r_{23} \neq r_{32}$, $t_{12} \neq t_{13}$, and $t_{21} \neq t_{31}$. The transmission of spin waves across the bilayer for a $\lambda_1 \rightarrow$

$\lambda_3$ mode conversion at the first interface (Supplementary Fig. 17a) can be written as

$$
\begin{aligned}
T &= t_{13}t_{31}e^{i|k_3|w} + t_{13}t_{31}r_{32}r_{23}e^{i(|k_2|+2|k_3|)w} \\
&\quad + t_{13}t_{31}r_{32}^2r_{23}^2e^{i(2|k_2|+3|k_3|)w} + \dots \\
&= t_{13}t_{31}e^{i|k_3|w}\sum_{j=0}^{\infty}r_{32}^jr_{23}^je^{ij(|k_2|+|k_3|)w} \\
&= t_{13}t_{31}e^{i|k_3|w}\frac{1}{1-r_{32}r_{23}e^{i(|k_2|+|k_3|)w}}.
\end{aligned}
\tag{5}
$$

Substitution of $t_{13}t_{31} = Ae^{i\varphi_A}$ and $r_{32}r_{23} = Be^{i\varphi_B}$, where $\varphi_A$ and $\varphi_B$ correspond to added phase shifts at the interfaces, gives

$$
\begin{aligned}
T &= \frac{Ae^{i(|k_3|w+\varphi_A)}}{1-Be^{i((|k_2|+|k_3|)w+\varphi_B)}} \\
&= \frac{A}{e^{-i(|k_3|w+\varphi_A)}-Be^{i(|k_2|w+\varphi_B-\varphi_A)}}.
\end{aligned}
\tag{6}
$$

Minimum transmission occurs when the denominator of Eq. (6) reaches a maximum value. For fixed reflection and transmission coefficients, this condition is satisfied when the phase of $e^{-i(|k_3|w+\varphi_A)}$ and $Be^{i(|k_2|w+\varphi_B-\varphi_A)}$ differ by an odd number of $\pi$. The condition of minimum transmission is therefore given by

$$
(|k_2|+|k_3|)w + \varphi_B = (2n+1)\times\pi.
\tag{7}
$$

This condition corresponds to destructive interference between the incoming $\lambda_3$ wave and waves propagating in the same direction after full cycles of internal reflection. It is noteworthy that Eq. (7) depends on $\varphi_B$ but not on $\varphi_A$. The resonance condition is thus fully determined by phase accumulation during spin-wave transport and reflections inside the YIG/CoFeB bilayer region and it is not affected by phase shifts incurred upon transmission into or out of the bilayer region. We note that Eq. (7) describing destructively interfering harmonic plane waves (in analogy to an optical Fabry–Pérot resonator) does not strictly apply to our magnonic device structure. Particularly, the long-range nature of dipolar interactions modifies the values of $k_1$, $k_2$, and $k_3$ near the interfaces, as outlined in ref. [15]. Moreover, dynamic dipolar coupling between the ferromagnetic metal stripe and the YIG film produces a frequency shift between the destructive interference conditions within the bilayer region and the transmission gaps. Micromagnetic simulations in Supplementary Fig. 10 illustrate this deviation from conventional Fabry–Pérot behavior. Both dipolar effects are taken into account in the model by adding a phase shift $\varphi_{dip}$ to each internal reflection cycle. Writing $\varphi_0 = \varphi_B + \varphi_{dip}$, the resonance condition producing minimum transmission is then given by

$$
(|k_2|+|k_3|)w + \varphi_0 = (2n+1)\times\pi.
\tag{8}
$$

We find that the frequencies of experimentally measured and simulated transmission gaps ($n = 1$–$18$) are well described by this resonance condition assuming a constant value of $\varphi_0 = 1.34\pi$ (Fig. 3c and Supplementary Fig. 9). The agreement demonstrates that the inclusion of dipolar effects via an effective phase shift is a good first-order approximation.

Equations (5)–(8) describe the scenario where an incoming $\lambda_1$ wave is converted to a short-wavelength $\lambda_3$ wave at the first interface. A similar set of equations can be derived for the configuration where an incoming $\lambda_1$ wave is converted to a $\lambda_2$ wave (Supplementary Fig. 17b). Importantly, the resonance condition for minimum transmission for this situation is also given by Eq. (8). Equation (5), however, takes the form

$$
T = t_{12}t_{21}e^{i|k_2|w}\frac{1}{1-r_{32}r_{23}e^{i(|k_2|+|k_3|)w}}.
\tag{9}
$$

Because the prefactors in Eqs. (5) and (9) are different, the spin-wave transmission signal at the gap frequency should be dissimilar for these two configurations. Nonreciprocity at the resonance condition is caused by stronger damping of $\lambda_3$ waves than $\lambda_2$ modes inside the bilayer and differences in the transmission coefficients $t_{13}$, $t_{31}$, $t_{12}$, and $t_{21}$. This nonreciprocal resonance effect is confirmed by the experimental data in Fig. 1b and Supplementary Fig. 4.

**Micromagnetic simulations**. We performed micromagnetic simulations using open-source GPU-accelerated MuMax3 software[25]. The YIG film thickness was 100 nm and its in-plane dimensions were set to 655 μm along $x$ and 0.64 μm along $y$. Two 50-nm-thick CoFeB stripes centered at $x = \pm 100$ μm were placed on top of the YIG film to simultaneously simulate spin-wave transmission for negative and positive wave vectors. The CoFeB stripes were separated from the YIG film by a 5-nm-thick nonmagnetic layer (accounting for the TaO$_x$ spacer in experiments). One-dimensional periodic boundary conditions were applied along the $y$ axis. We discretized the simulation area into $10 \times 10 \times 5$ nm$^3$ cells. Spin waves were excited by applying a sinusoidal or sinc-function-type magnetic field across a 2-μm-wide area at $x = 0$. The excitation field was 0.05 mT. A magnetic bias field was applied along the y axis. As input parameters, we used saturation magnetizations $M_{s,YIG} = 1.92 \times 10^5$ A/m and $M_{s,CoFeB} = 1.15 \times 10^6$ A/m, exchange constants $A_{ex,YIG} = 3.1 \times 10^{-12}$ J/m and $A_{ex,CoFeB} = 1.6 \times 10^{-11}$ J/m, and magnetic damping parameters $\alpha_{YIG} = 0.001$ and $\alpha_{CoFeB} = 0.005$. We intentionally used a relatively large damping parameter for YIG to limit the computation time and reduce reflections from the edges of the simulation area. During continuous excitation, the time evolutions of

the $x$-component ($m_x$) and $z$-component ($m_z$) of magnetization were recorded for 200 ns. Spatially resolved spin-wave intensity maps were obtained by Fourier-transforming the evolutions of $m_x$ or $m_z$ on a cell-by-cell basis.

**Reporting summary**. Further information on research design is available in the Nature Research Reporting Summary linked to this article.

## Data availability
All data that support the findings of this study are available from the corresponding authors upon reasonable request. Source data are provided with this paper.

## Code availability
Wolfram Mathematica 12 was used to calculate the bilayer dispersion curves. The code is available from the corresponding authors on reasonable request.

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

## Acknowledgements

This work was supported by the Academy of Finland (Grant Nos. 317918, 316857, 321983 and 325480) and the German Research Foundation (DFG) via CRC 227 and SPP 2137. Lithography was performed at the Micronova Nanofabrication Centre, supported by Aalto University. Computational resources were provided by the Aalto Science-IT project.

## Author contributions

H.J.Q. and S.v.D. conceived and supervised the research. H.J.Q. and F.H. fabricated the samples and performed the broadband spin-wave spectroscopy measurements. R.B.H. conducted the analytical calculations. H.J.Q. and R.B.H. performed the micromagnetic simulations. R.D. and G.W. conducted the SNS-MOKE microscopy measurements. L.F. performed the TR-MOKE microscopy experiments. R.B.H., H.J.Q., and S.v.D. wrote the manuscript. All authors discussed the results.

## Competing interests

The authors declare no competing interests.
