## [Peer Review File · Nature Communications]

Reviewers' Comments:

Reviewer #1:

Remarks to the Author:

The manuscript "Nanoscale magnonic Fabry-Pérot resonator for low-loss spin-wave manipulation" by H. Qin et al. presents very interesting experimental results on the magnonic counterpart of the optical Fabry-Pérot resonator. Spin waves or magnons act as promising candidates for the next generation of information carriers with low power consumptions. In this work, the authors demonstrate a simple and smart method to realize a magnonic Fabry-Pérot resonator by placing a single ferromagnetic nanowire on top of a YIG film. The explanation is based on the asymmetric dispersion due to the dipolar coupling between the ferromagnetic nanowire and YIG film. The micromagnetic simulations agree well with the experimental results using VNA and MOKE techniques. All in all, the presented work is new and quite valuable for the magnonic community and beyond, and hence I believe that the results can be suitable for a publication in Nature Communications after revision.

Major issues to be considered:

1. While this work is a mimic of the Fabry-Pérot resonator, I find the discussion and comparison between two different subjects (magnonics and optic Fabry-Pérot resonator) is not sufficient in the manuscript. For example, the entire introduction does not mention Fabry-Pérot resonator while the introduction of magnonic crystals is over-sufficient. I don't think magnonic crystals have a strong connection with this work. Instead, I suggest the authors should introduce the comparison between Fabry-Pérot resonator in optics and magnonics.
2. The main results of this work are based on the low-loss YIG thin film. The authors may want to cite the pioneering works on high-quality thin-film YIG, such as IEEE Magn. Lett. 5, 6700104 (2014), Appl. Phys. Lett. 103, 082408 (2013) and Sci. Rep. 4, 6848 (2014).
3. In page 7, the sentence "in the following, we'll demonstrate that the transmission gaps are caused by destructive interference..., analogous to the operation of an optical Fabry-Pérot resonator." is written in such an elegant manner and reveals the key physics the article try to investigate, but unfortunately it comes rather late in the text... I suggest that the authors first introduce conceptually this essential physics and then go into details of the descriptions of sample structures and measurement techniques of VNA and MOKE.
4. The authors claim that the dynamic dipolar coupling between the CoFeB nanowire and YIG film is the main reason for the wavelength conversion. However, at the spin-wave frequency in YIG, CoFeB nanowires are clearly off-resonance due to the high saturation magnetization. I would guess that the static dipolar field generated by the CoFeB leads to the asymmetric spin wave dispersion.
5. In this work, the authors observe short spin-wave wavelength in the CoFeB nanowire covered area, which is quite impressive, because one can excite spin waves with wavelengths much shorter than the nanowire width. Meanwhile, I suggest the authors to refer to other methods for exciting short-wavelength spin waves such as STNO and nano-grating couplers.
6. The calculation of the asymmetric dispersion relation is very important for understanding the experimental results. The authors may consider to provide some theoretical analysis in the main text, rather than all in the Method.
7. In the destructive interference condition, when the spin waves within the bilayer accumulate a total phase of $(2n+1)\pi$, the spin waves cannot pass through the nanowire region. The readers may wonder where the energy go for the incoming spin waves. Can the authors comment on this?
8. The active control of the spin wave spectra with P or AP state is indeed significant for the reconfigurable magnonics. If possible, the authors may want to present angle-dependent measurements to reveal how the magnonic resonator behaviors gradually change from P state to AP state.

Minor issues or suggestions:

1. In the second sentence in the abstract, The term of "low-loss transporting material" is somehow misleading for people working on transport measurement which refers to electron transport by default. I find "transporting material" is an odd terminology... The authors perhaps want to express something like "yttrium iron garnet (YIG) is the prime candidate for spin-wave transport with low

loss."

2. At the beginning of page 3, "designer magnonics" may be a typo and should perhaps be "designed magnonics".
3. The English may still be improved. Some redundant words may be removed to keep the text concise, e.g. Page 4, "physically separated" can well be just "separated", "spacer layer" can be simply "spacer". Also, "We consider" may mean "We use" or "We adapt". These are all details, but I believe some more improvement on the language may further improve the readability and the impact of the paper.
4. Fig. 1b, The subscripts of S12 and S21 are too small to read in a printed version, which I suggest to be moved to the right corner of the sub-figures, since the upper panel is S12 and the bottom panel is S21...
5. In Figs. 1e to h, the combination of both experimental and simulation results is too crowded and I cannot get key features from these figures. The authors may want to separate Fig.1 into two figures...
6. The imaginary parts of the VNA results should be presented as the authors done in their earlier paper (Nat. Commun. 9, 5445 (2018)), in order to reveal the phase information of the propagating spin waves.

Reviewer #2:

None

Reviewer #3:

Remarks to the Author:

In the manuscript the authors report measurements and modelling of a magnonic system composed of a plain film of the insulator Yttrium-Iron-Garnet (YIG) with a strip of metallic ferromagnet on top (mainly CoFeB and as a reference, also Py) which is dipolar coupled to the YIG. They show that in this system, several effects occur which they explained by the strongly shifted and non-reciprocal dispersion relation in the bilayer region. The most interesting effects in my opinion is the appearance of a resonance dip in the transmitted SW signal intensity which can be caused by a single CoFeB strip whose width is significantly smaller compared to the wavelength of the spin waves the bare YIG film. The resonance dip in the transmission is somehow similar to the well-known findings of Ref. 31 of Kostylev et al. from 2007 but in the current work, the explanation using the Fabry Perot modes is much more intuitive and the use of a second magnetic material add the non-reciprocity of transmission, which is absent in Ref. 31. In addition, it is exciting to see that the spin-wave transmission outside the resonance is almost not affected and that the resonance can be comparably easily tuned by changing the width of the strips and their number/material etc. A hysteric reconfiguration is also shown where the authors tune the resonance frequency by changing the coupling between the CoFeB strip and the YIG via a parallel or anti-parallel alignment.

I found the paper interesting and in general, well written. The results are, in general, credible and the resonance effect is also well explained.

However, I have some issues which I think should be addressed before a decision on a potential publication in such a high-level journal like nature communications can be made:.

1)The authors write: "Away from the transmission gaps, transport of spin waves across the YIG/CoFeB bilayer matches that of the uncovered YIG film, irrespective of the propagation direction or the orientation of magnetic field."

This is true and for me, this is one of the quite surprising findings of this manuscript. Since the low transmission losses outside the resonance condition are a broadband effect, I think that they should be related to generally very high transmission coefficients like t_{12} etc. Given the large change in wavelength at the edges of the resonator, I found this somehow surprising since it means in an optically analogy that the "impedance matching" is very good for a very broad range

of wavelengths. Unfortunately, I could not find any values for the reflection and transmission coefficients in the manuscript. So my request would be to at least estimate those values from the reconstruction of the experimental data with the analytical model or micromagnetic simulations.

If one has a look on literature concerning the transmission of spin waves at boundaries separating two regions with different dispersion relations there is a reference of M. Kostylev which is dealing with a somehow similar system: Phys Rev B, vol. 102, 014445, (2020). From what can be concluded from this Ref and from the literature cited there (Phys. Rev. Lett. 122, 197201 and Phys. Rev. B 101, 144430) the reflection at a single boundary is very interesting by itself and quite strongly depending on the mode type (MSSW or BVMSW). Thus, it is needed to compare the finding of the manuscript under review with the predictions in these manuscripts. Did the authors try to study the system also in BVMSW configuration?

Concerning the explanation of the non-reciprocity of the S_{21} and S_{12} transmission parameters, I agree with the qualitative interpretation by the authors that the two different wavelength in the resonator have different decay length. However, I am surprised that similar to the reflection coefficients, there are no values given for the decay lengths of the different modes? It should be possible to at least estimate the decay lengths and their ratio using the obtained dispersion relations and compare the values to the experimental ratio of S_{21}/S_{12} .

2) The authors write „Because of the asymmetric dispersion relation in the bilayer region, r_{23} not equal r_{32} “ etc

It is not clear to me if a non-reciprocity of the dispersion relation is always causing this non-reciprocity of the reflection coefficients. Is this a general effect which also occurs, e.g., for a non-reciprocal magnonic systems with DMI?

3) The authors mention a low Gilbert damping of their YIG film. However, the "inhomogeneous linewidth" (Δf extrapolated for $H=0$) is not given and seems to be significant. Could the authors please discuss the contribution of the additional broadening to Δf for the used frequencies and the potential effect on the decay length of the spin waves? Have the decay lengths been measured, maybe via TR-MOKE?

Reviewer #1

Reviewer comment

The manuscript “Nanoscale magnonic Fabry-Pérot resonator for low-loss spin-wave manipulation” by H. Qin et al. presents very interesting experimental results on the magnonic counterpart of the optical Fabry-Pérot resonator. Spin waves or magnons act as promising candidates for the next generation of information carriers with low power consumptions. In this work, the authors demonstrate a simple and smart method to realize a magnonic Fabry-Pérot resonator by placing a single ferromagnetic nanowire on top of a YIG film. The explanation is based on the asymmetric dispersion due to the dipolar coupling between the ferromagnetic nanowire and YIG film. The micromagnetic simulations agree well with the experimental results using VNA and MOKE techniques. All in all, the presented work is new and quite valuable for the magnonic community and beyond, and hence I believe that the results can be suitable for a publication in Nature Communications after revision.

Our response

We would like to thank the reviewer for the positive assessment of our work. Below, we respond to the reviewer’s remarks on a point-by-point basis. In the manuscript, revisions are highlighted in red.

Reviewer comment

Major issues to be considered:

1. While this work is a mimic of the Fabry-Pérot resonator, I find the discussion and comparison between two different subjects (magnonics and optic Fabry-Pérot resonator) is not sufficient in the manuscript. For example, the entire introduction does not mention Fabry-Pérot resonator while the introduction of magnonic crystals is over-sufficient. I don’t think magnonic crystals have a strong connection with this work. Instead, I suggest the authors should introduce the comparison between Fabry-Pérot resonator in optics and magnonics.

Our response

We revised the introduction of the manuscript. The optical Fabry-Pérot resonator is described and requirements for a magnonic analog are stipulated.

Reviewer comment

2. The main results of this work are based on the low-loss YIG thin film. The authors may want to cite the pioneering works on high-quality thin-film YIG, such as IEEE Magn. Lett. 5, 6700104 (2014), Appl. Phys. Lett. 103, 082408 (2013) and Sci. Rep. 4, 6848 (2014).

Our response

We added the references to works on high-quality thin-film YIG to the introduction of the manuscript.

Reviewer comment

3. In page 7, the sentence “in the following, we’ll demonstrate that the transmission gaps are caused by destructive interference..., analogous to the operation of an optical Fabry-Pérot resonator.” is written in such an elegant manner and reveals the key physics the article try to investigate, but

unfortunately it comes rather late in the text... I suggest that the authors first introduce conceptually this essential physics and then go into details of the descriptions of sample structures and measurement techniques of VNA and MOKE.

Our response

The key physics that we investigate is now introduced on page 2 of the manuscript, well before the description of the sample structure and measurements techniques. Moreover, the aim of the paper, i.e., to demonstrate a magnonic analog to the optical Fabry-Pérot resonator, is now described at the beginning of the paper.

Reviewer comment

4. The authors claim that the dynamic dipolar coupling between the CoFeB nanowire and YIG film is the main reason for the wavelength conversion. However, at the spin-wave frequency in YIG, CoFeB nanowires are clearly off-resonance due to the high saturation magnetization. I would guess that the static dipolar field generated by the CoFeB leads to the asymmetric spin wave dispersion.

Our response

In our experiments, the CoFeB stripes are long and an external magnetic bias field aligns the magnetization in YIG and CoFeB parallel to the stripes (Damon-Eshbach geometry). In this configuration, the CoFeB stripes do not produce a static dipolar field. Instead, the incoming spin waves in the YIG film induce spin waves of equal wavelength in the CoFeB stripes via dynamic dipolar coupling. Since the interaction occurs well below the FMR frequency of CoFeB, the induced magnetization dynamics in CoFeB is driven/forced. A new Supplementary Fig. 7 demonstrates the excitation of spin wave in CoFeB stripes on top of a YIG film. The asymmetric spin wave dispersion of the YIG/CoFeB bilayer, which we calculate by extending Peter Grünberg's model, is produced by dynamic dipolar coupling between propagating spin-wave modes (see Methods).

Reviewer comment

5. In this work, the authors observe short spin-wave wavelength in the CoFeB nanowire covered area, which is quite impressive, because one can excite spin waves with wavelengths much shorter than the nanowire width. Meanwhile, I suggest the authors to refer to other methods for exciting short-wavelength spin waves such as STNO and nano-grating couplers.

Our response

We added references to other methods for exciting short-wavelength spin waves on page 7 of the manuscript.

Reviewer comment

6. The calculation of the asymmetric dispersion relation is very important for understanding the experimental results. The authors may consider to provide some theoretical analysis in the main text, rather than all in the Method.

Our response

We carefully considered the reviewer's suggestion. In our work we use Peter Grünberg's model (J. Appl. Phys. 52, 6824 (1981)) to calculate the asymmetric dispersion relation of the magnetic bilayer. To correctly calculate the dispersion relation for the short-wavelength λ_3 mode, we extended the model by adding exchange terms for the individual CoFeB and YIG layers. The substitution that we used to implement this modification is now indicated in the main text. Providing further details on a previously published model in the main text would, in our opinion, distract from the discussion of experimental results.

Reviewer comment

7. In the destructive interference condition, when the spin waves within the bilayer accumulate a total phase of $(2n+1)\pi$, the spin waves cannot pass through the nanowire region. The readers may wonder where the energy go for the incoming spin waves. Can the authors comment on this?

Our response

For CoFeB nanostripes, the resonator reflects spin waves with almost no energy loss when the destructive interference condition is met. To demonstrate this, we added Supplementary Fig. 8 showing phase-resolved TR-MOKE microscopy maps and calculations based on a three-wave reflection/transmission model at five different frequencies. For a 730-nm-wide CoFeB stripe, we only obtain good agreement between the experimental data and calculations when the absorption coefficient is set close to zero, demonstrating low-loss spin-wave transport across the resonator. For broad CoFeB stripes, some energy is absorbed by dynamic dipolar coupling to higher-loss CoFeB inside the resonator. We discuss the new Supplementary Fig. 8 on page 10 of the manuscript.

Reviewer comment

8. The active control of the spin wave spectra with P or AP state is indeed significant for the reconfigurable magnonics. If possible, the authors may want to present angle-dependent measurements to reveal how the magnonic resonator behaviors gradually change from P state to AP state.

Our response

We followed the reviewer's suggestion and performed angular-dependent measurements on a resonator structure using broadband spin-wave spectroscopy. The new results are included as a new Supplementary Fig. 16. In the experiments, we recorded the S12 scattering parameter while rotating a 10 mT magnetic bias field by 360° within the sample plane. At this field strength, the magnetization of the CoFeB stripe remains fixed, whereas the magnetization of the YIG films aligns along the external bias field. The result demonstrates continuous tuning of the transmission gap frequency upon magnetization rotation away from the DE configuration. We do not measure a transmission signal when the angle between the wave vector and the magnetization of the YIG film is small, because of inefficient spin-wave excitation by the microwave antenna.

Reviewer comment

Minor issues or suggestions:

1. In the second sentence in the abstract, The term of "low-loss transporting material" is somehow misleading for people working on transport measurement which refers to electron transport by

default. I find “transporting material” is an odd terminology... The authors perhaps want to express something like “yttrium iron garnet (YIG) is the prime candidate for spin-wave transport with low loss.”

Our response

We changed the terminology to the one suggested by the reviewer.

Reviewer comment

2. At the beginning of page 3, “designer magnonics” may be a typo and should perhaps be “designed magnonics”.

Our response

We removed the term “designer magnonics”.

Reviewer comment

3. The English may still be improved. Some redundant words may be removed to keep the text concise, e.g. Page 4, “physically separated” can well be just “separated”, “spacer layer” can be simply “spacer”. Also, “We consider” may mean “We use” or “We adapt”. These are all details, but I believe some more improvement on the language may further improve the readability and the impact of the paper.

Our response

Following the reviewer’s suggestion, we polished the English and removed redundant words.

Reviewer comment

4. Fig. 1b, The subscripts of S12 and S21 are too small to read in a printed version, which I suggest to be moved to the right corner of the sub-figures, since the upper panel is S12 and the bottom panel is S21...

Our response

We moved the labels S12 and S21 from the figure legend to the y-axis label for better readability.

Reviewer comment

5. In Figs. 1e to h, the combination of both experimental and simulation results is too crowded and I cannot get key features from these figures. The authors may want to separate Fig.1 into two figures...

Our response

We followed the reviewer’s suggestion and separated Fig. 1 into two figures.

Reviewer comment

6. The imaginary parts of the VNA results should be presented as the authors done in their earlier paper (Nat. Commun. 9, 5445 (2018)), in order to reveal the phase information of the propagating spin waves.

Our response

We added the real and imaginary parts of the data shown in Fig. 1b to Supplementary Fig. 3

Reviewer #3

Reviewer comment

In the manuscript the authors report measurements and modelling of a magnonic system composed of a plain film of the insulator Yttrium-Iron-Garnet (YIG) with a strip of metallic ferromagnet on top (mainly CoFeB and as a reference, also Py) which is dipolar coupled to the YIG. They show that in this system, several effects occur which they explained by the strongly shifted and non-reciprocal dispersion relation in the bilayer region. The most interesting effects in my opinion is the appearance of a resonance dip in the transmitted SW signal intensity which can be caused by a single CoFeB strip whose width is significantly smaller compared to the wavelength of the spin waves the bare YIG film. The resonance dip in the transmission is somehow similar to the well-known findings of Ref. 31 of Kostylev et al. from 2007 but in the current work, the explanation using the Fabry Perot modes is much more intuitive and the use of a second magnetic material adds the non-reciprocity of transmission, which is absent in Ref. 31. In addition, it is exciting to see that the spin-wave transmission outside the resonance is almost not affected and that the resonance can be comparably easily tuned by changing the width of the strips and their number/material etc. A hysteric reconfiguration is also shown where the authors tune the resonance frequency by changing the coupling between the CoFeB strip and the YIG via a parallel or anti-parallel alignment.

I found the paper interesting and in general, well written. The results are, in general, credible and the resonance effect is also well explained. However, I have some issues which I think should be addressed before a decision on a potential publication in such a high-level journal like nature communications can be made:

Our response

We would like to thank the reviewer for the positive assessment of our work. Below, we respond to the reviewer's remarks on a point-by-point basis. In the manuscript, revisions are highlighted in red.

Reviewer comment

1)The authors write: "Away from the transmission gaps, transport of spin waves across the YIG/CoFeB bilayer matches that of the uncovered YIG film, irrespective of the propagation direction or the orientation of magnetic field." This is true and for me, this is one of the quite surprising findings of this manuscript. Since the low transmission losses outside the resonance condition are a broadband effect, I think that they should be related to generally very high transmission coefficients like t_{12} etc. Given the large change in wavelength at the edges of the resonator, I found this somehow surprising since it means in an optically analogy that the "impedance matching" is very good for a very broad range of wavelengths. Unfortunately, I could not find any values for the

refection and transmission coefficients in the manuscript. So my request would be to at least estimate those values from the reconstruction of the experimental data with the analytical model or micromagnetic simulations.

If one has a look on literature concerning the transmission of spin waves at boundaries separating two regions with different dispersion relations there is a reference of M. Kostylev which is dealing with a somehow similar system: Phys Rev B, vol. 102, 014445, (2020). From what can be concluded from this Ref and from the literature cited there (Phys. Rev. Lett. 122, 197201 and Phys. Rev. B 101, 144430) the reflection at a single boundary is very interesting by itself and quite strongly depending on the mode type (MSSW or BVMSW). Thus, it is needed to compare the finding of the manuscript under review with the predictions in these manuscripts. Did the authors try to study the system also in BVMSW configuration?

Concerning the explanation of the non-reciprocity of the S21 and S12 transmission parameters, I agree with the qualitative interpretation by the authors that the two different wavelength in the resonator have different decay length. However, I am surprised that similar to the reflection coefficients, there are no values given for the decay lengths of the different modes? It should be possible to at least estimate the decay lengths and their ratio using the obtained dispersion relations and compare the values to the experimental ratio of S21/S12.

Our response

We made several revisions to the manuscript to address the reviewer's comments. Using micromagnetic simulations (new Supplementary Fig. 7), we now analyze the transmission coefficients of the resonator interfaces. For spin waves converting from the λ_1 to the λ_2 mode (incoming waves) or vice versa (outgoing waves), we estimate a transmission coefficient $t_{12} = 1$ and $t_{21} = 0.5$, respectively. Quantitative assessments of transmissions involving λ_1 and λ_3 modes are complicated by the differences in wave localization and spin-wave ellipticity ($\varepsilon = 1 - m_{\min}/m_{\max}$). While the wave profiles of the λ_1 and λ_2 modes are approximately uniform across the YIG film thickness, the short-wavelength λ_3 mode localizes at the top surface. Also, λ_1 and λ_2 waves are elliptical along x ($\varepsilon = 0.8$), whereas the λ_3 mode near the top surface is slightly elliptical along z ($\varepsilon = 0.3$). A discussion on the values of transmission coefficients is now included on page 9-10 of the manuscript. Importantly, we note that spin-wave reflection at single interfaces does not contradict low-loss spin-wave transmission across nanoresonators at allowed frequencies, as presented in Fig. 1. Analogous to an optical Fabry-Pérot resonator, multiple internal reflections and constructive interference between incoming and circulating spin waves produces high transmission even if the reflection coefficients of the resonator interfaces are non-zero.

In the new text on spin-wave transmission across single boundaries, we included references to the papers mentioned by the reviewer. The article by Verba et al. (Phys. Rev. B 101, 144430 (2020)) is included as it discusses how spin-wave ellipticity affects spin-wave transmission through an internal boundary. The papers by Mohseni and coworkers (Phys. Rev. Lett. 122, 197201 (2019) and Phys. Rev. B 102, 014445, (2020)) demonstrate backscattering immunity for propagating spin-waves in the Damon-Eshbach configuration, which may play an important role in obtaining small reflection coefficients for some mode conversions in our experiments.

In the revised manuscript we include new experimental data obtained while rotating the YIG magnetization by 360° within the sample plane (Supplementary Fig. 16). The result demonstrates

continuous tuning of the transmission gap frequency upon magnetization rotation away from the Damon-Eshbach configuration. We do not measure a transmission signal when the angle between the wave vector and the magnetization of the YIG film is small (i.e. in the BVMSW configuration), because of inefficient spin-wave excitation by the microwave antenna.

We extracted the decay length of all three spin-wave modes in our experiments. The decay length of the λ_1 and λ_2 modes are extracted by fitting TR-MOKE line profiles (new Supplementary Fig. 6). The short-wavelength λ_3 mode is derived from a fit to the nonreciprocity coefficient ($|S_{21} - S_{12}|/|S_{21} + S_{12}|$) in Supplementary Fig. 5. The decay lengths that we directly derived from the TR-MOKE experiments agree with those calculated using the obtained dispersion relations (see response to last comment). As an example of data consistency, we consider the S_{21}/S_{12} ratio, which is given by $\exp(-x/l_d(\lambda_2))/\exp(-x/l_d(\lambda_3))$. Here, $l_d(\lambda_2)/l_d(\lambda_3)$ is given by the group-velocity ratio $v_g(\lambda_2)/v_g(\lambda_3)$. Taking $f = 1.58$ GHz as an example, we derive $v_g(\lambda_2)/v_g(\lambda_3) \approx 4$ from the dispersion relation (see Fig. 3a), i.e., $l_d(\lambda_2) \approx 4l_d(\lambda_3)$. Using this data and setting the CoFeB stripe width to $30 \mu\text{m}$ gives $S_{21}/S_{12} \approx 20$ at $f = 1.58$ GHz, which agrees well to the S_{12} and S_{21} data shown in Supplementary Fig. 5. The extracted decay lengths are discussed on page 7 of the main manuscript.

Reviewer comment

2) The authors write „Because of the asymmetric dispersion relation in the bilayer region, r_{23} not equal r_{32} ” etc

It is not clear to me if a non-reciprocity of the dispersion relation is always causing this non-reciprocity of the reflection coefficients. Is this a general effect which also occurs, e.g., for a non-reciprocal magnonic systems with DMI?

Our response

In the revised manuscript we discuss the reflection and transmission coefficient in more detail and we point out that differences in wave localization and spin-wave ellipticity may play a major role. It is therefore not only the non-reciprocity of the dispersion relation that causes non-reciprocity of the reflection and transmission coefficients. To acknowledge this, we changed the sentence to: “Because of the asymmetric dispersion relation in the bilayer region and differences in wave localization and spin-wave ellipticity (Supplementary Fig. 7), ...”

Reviewer comment

3) The authors mention a low Gilbert damping of their YIG film. However, the "inhomogeneous linewidth" (Δf extrapolated for $H=0$) is not given and seems to be significant. Could the authors please discuss the contribution of the additional broadening to Δf for the used frequencies and the potential effect on the decay length of the spin waves? Have the decay lengths been measured, maybe via TR-MOKE?

Our response

The inhomogeneous linewidth (Δf extrapolated for $H=0$) is 17 MHz. This relatively large linewidth can be attributed to the distribution of wave vectors that our coplanar waveguides excite and some magnetic inhomogeneities in the YIG film. These effects, however, do not influence the decay length of the spin waves much. Using a magnetic damping parameter $\alpha = 5 \times 10^{-4}$ (derived

from data in Fig. S1b) and $l_d = v_g/(2\pi\alpha f)$, with the group velocity v_g derived from the dispersion relations in Fig. 3a, we find $l_d = 630 \mu\text{m}$ for λ_1 , $l_d = 32 \mu\text{m}$ for λ_2 , and $l_d = 10 \mu\text{m}$ for λ_3 at 1.8 GHz. These decay lengths are comparable to those derived from TR-MOKE microscopy measurements (see new data in Supplementary Figs. 5 and 6).

Reviewers' Comments:

Reviewer #1:

Remarks to the Author:

The authors have very well addressed all my comments and made updates on the manuscript accordingly. It is appreciated that some additional experiments were conducted such as angle-dependent measurements that have in my opinion strengthened the argument of the paper substantially.

In particular, I am now convinced that the effect of dynamic dipolar coupling between YIG and CoFeB could be responsible for the experimental observation. Although the coupling occurs not directly at the FMR of CoFeB, this process may be triggered by an "evanescent-type" magnon-magnon coupling mediated by dynamic dipolar coupling between YIG and CoFeB, which is similarly referred by the authors as "driven/forced". In this case, the authors may consider to additionally mention some pioneering works on dynamic dipolar coupling, such as Pigeau, B. et al. Phys. Rev. Lett. 109, 247602 (2012).

All in all, I recommend the publication of this work in Nature Communications after considering the minor comment/suggestion above.

Reviewer #3:

Remarks to the Author:

The authors have addressed all my comments and added extensive additional material to the supplement which is giving a lot of details. This enhances the understanding of the systems details for the interested reader significantly. I can recommend the manuscript for publication nature communications without further modifications.

Reviewer #1

Reviewer comment

The authors have very well addressed all my comments and made updates on the manuscript accordingly. It is appreciated that some additional experiments were conducted such as angle-dependent measurements that have in my opinion strengthened the argument of the paper substantially. In particular, I am now convinced that the effect of dynamic dipolar coupling between YIG and CoFeB could be responsible for the experimental observation. Although the coupling occurs not directly at the FMR of CoFeB, this process may be triggered by an "evanescent-type" magnon-magnon coupling mediated by dynamic dipolar coupling between YIG and CoFeB, which is similarly referred by the authors as "driven/forced". In this case, the authors may consider to additionally mention some pioneering works on dynamic dipolar coupling, such as Pigeau, B. et al. Phys. Rev. Lett. 109, 247602 (2012). All in all, I recommend the publication of this work in Nature Communications after considering the minor comment/suggestion above.

Our response

We would like to thank the reviewer for his/her recommendation to publish our work in Nature Communications. The suggested reference has been added to the manuscript.

Reviewer #3

Reviewer comment

The authors have addressed all my comments and added extensive additional material to the supplement which is giving a lot of details. This enhances the understanding of the systems details for the interested reader significantly. I can recommend the manuscript for publication nature communications without further modifications.

Our response

We would like to thank the reviewer for his/her recommendation to publish our work in Nature Communications.